# FAST VISUOMOTOR POLICY FOR ROBOTIC MANIPULATION

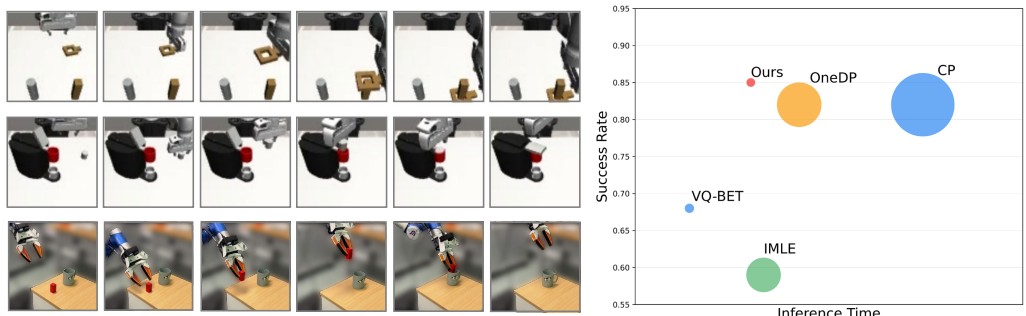

Figure 1: **Left:** Our method achieves strong results in three settings: single-task (top), multi-task (middle), and real-world (bottom). Each row displays rollouts from representative tasks within each scenario. **Right:** The bubble chart illustrates a representative comparison on PushT (Chi et al., 2023), showing that our method achieves state-of-the-art success rates while being faster in inference than the strongest baselines. Bubble sizes indicate model parameter counts, demonstrating that our approach delivers competitive performance with significantly smaller models.

## ABSTRACT

We present a fast and effective policy framework for robotic manipulation, named **Energy Policy**, designed for high-frequency robotic tasks and resource-constrained systems. Unlike existing robotic policies, Energy Policy natively predicts multimodal actions in a single forward pass, enabling high-precision manipulation at high speed. The framework is built upon two core components. First, we adopt the energy score as the learning objective to facilitate multimodal action modeling. Second, we introduce an energy MLP to implement the proposed objective while keeping the architecture simple and efficient. We conduct comprehensive experiments in both simulated environments and real-world robotic tasks to evaluate the effectiveness of Energy Policy. The results show that Energy Policy matches or surpasses the performance of state-of-the-art manipulation methods while significantly reducing computational overhead. Notably, on the MimicGen benchmark, Energy Policy achieves superior performance with at a faster inference compared to existing approaches.

## 1 INTRODUCTION

Policy learning from demonstrations has emerged as a powerful paradigm for enabling robots to acquire complex skills. It is typically formulated as a supervised regression task, where observations are mapped to actions. To achieve high-precision action regression, incorporating generative models into policy learning has become a dominant approach across various robotic tasks.

Recent research has aimed to enhance policy learning by introducing various generative modeling techniques. Adopting Autoregressive Modeling (AM) from large language models (Kim et al., 2024; Brohan et al., 2022; 2023; Qu et al., 2025) has proven to be a powerful solution, owing to its scalability, flexibility, and mature exploration. However, like language models, AM uses discrete action tokens for action prediction, which can sacrifice fine-grained action details. Recently, there has

been considerable research into continuous action representations. While directly applying L1 or L2 regression (Kim et al., 2025; Su et al., 2025) to continuous action prediction offers a straightforward way to improve action precision, it struggles with multimodal action distributions due to its uni-modal modeling approach. Diffusion Modeling (DM) (Chi et al., 2023; Ze et al., 2024a; Wang et al., 2024b; Team et al., 2024; Black et al., 2024; Intelligence et al., 2025) provides a promising alternative by learning multimodal distributions through modeling the gradient of the action score function. However, DM requires multiple denoising steps, making it computationally prohibitive for real-time robotic tasks.

In this paper, we propose a novel and efficient approach to policy learning that natively predicts multimodal continuous actions in a single forward pass. Specifically, during training, we utilize energy score (Brier, 1950; Gneiting & Raftery, 2007b) as the learning objective to minimize the distributional difference between the predicted actions and the ground truth. The energy score provides a rigorous measure of whether predictions match the underlying distribution, making it a natural choice for multimodal action modeling. To fully exploit this objective, we propose an energy MLP, a dedicated module that explicitly parameterizes energy score modeling. This design is central to our method, as it enhances representational expressiveness, allowing the energy score to serve as an effective supervisory signal for complex multimodal distributions. During inference, we can directly sample continuous actions from the model's distribution prediction, avoiding the need for multiple forward passes as in Diffusion Modeling. In addition, we incorporate parallel decoding, which generates all actions simultaneously and enables efficient action chunking (Zhao et al., 2023).

We conduct extensive experiments to demonstrate the effectiveness of our proposed method. Across a range of simulated robotic manipulation benchmarks, such as Robomimic (Mandlekar et al., 2021) and MimicGen (Mandlekar et al., 2023), our method achieves high task success rates and fast inference speeds. Notably, it outperforms CARP (Gong et al., 2024) across all benchmarks, with a $2.3\times \sim 7\times$ faster inference speed, and further surpasses existing efficient policies on the PushT task. We also evaluate our approach on real-world tasks under compute-constrained conditions. Compared to baseline methods, our method exhibits a higher success rate and faster inference, underscoring its suitability for real-time robotic applications.

In summary, our contributions are as follows: First, we present a novel approach to policy learning that models multimodal continuous actions. Second, the proposed method offers faster inference speeds, making it suitable for real-time robotic tasks. Third, extensive experiments validate the effectiveness of our method in both simulated and real-world robotic manipulation tasks.

## 2 RELATED WORK

**Learning Robotic Manipulation from Demonstrations.** Imitation learning enables robots to learn to perform tasks demonstrated by experts. Recently, there are various approaches to be developed for policy learning with different task constraints and control modalities. Autoregressive Modeling (AM) (Kim et al., 2024; Brohan et al., 2022; 2023; Qu et al., 2025) provides next-token prediction paradigm and use discrete action representation for manipulation learning. RT2 (Brohan et al., 2023) takes language instructions and visual observations as input, and outputs discrete action tokens in an auto-regressive manner. For high precision manipulation, enormous works (Kim et al., 2025; Su et al., 2025; Chi et al., 2023; Ze et al., 2024a; Wang et al., 2024b; Team et al., 2024; Black et al., 2024; Intelligence et al., 2025) explore continuous action representations. (Kim et al., 2025; Su et al., 2025) applies L1 or L2 objectives to learn to predict continuous action. However, these methods struggle with multimodal action distributions due to their uni-modal nature. Diffusion Policy (Chi et al., 2023) is proposed to handle multimodal action distributions by adopting a conditional denoising diffusion process, which involves multiple denoising steps. Unlike existing works, our method employs energy score to learn to predict continuous multimodal action.

**Fast Visuomotor Policy Learning.** In addition to manipulation precision, inference speed is another critical aspect of robotic policy. For AM-based models, integrating the KV-Cache technique can significantly enhance inference speed. Recent work such as Fast (Pertsch et al., 2025) introduces compressed action tokens to further improve runtime, while CARP (Gong et al., 2024) employs a next-scale autoregressive paradigm to shorten prediction horizons. Diffusion-based approaches typically rely on action chunking (Zhao et al., 2023) to achieve higher action throughput, and can

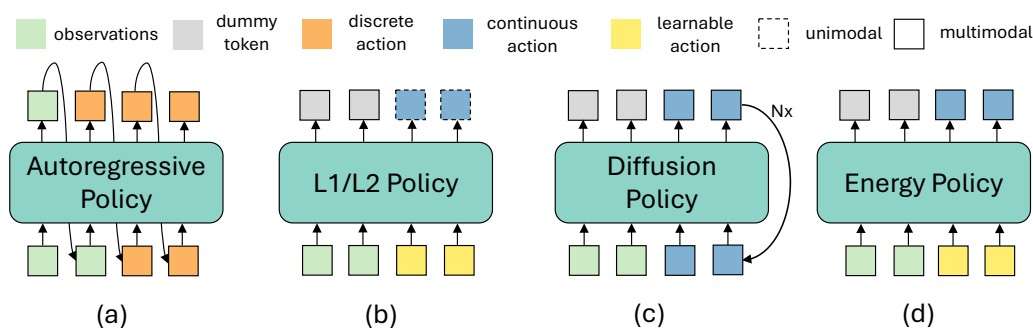

Figure 2: **Comparisons of existing Polies.** (a) Autoregressive policy predicts discrete tokens in an autoregressive manner. (b) L1/L2 policy predicts continuous actions but struggles with multimodal distribution modeling. (c) Diffusion policy generates multimodal continuous actions through multiple denoising steps. (d) Our Energy Policy produces multimodal continuous actions in a single forward pass.

leverage distillation techniques to reduce denoising steps (Prasad et al., 2024; Wang et al., 2024c), though often at the cost of action accuracy. Unlike these diffusion-based pipelines and their distilled variants, our approach natively predicts continuous actions in a single forward pass. This fundamental distinction eliminates the need for iterative refinement and avoids accuracy-compromising distillation, enabling our approach to achieve both high precision and fast inference simultaneously.

## 3 METHOD

In this section, we start by focusing on preliminaries, including problem formulation and existing works. Then we propose an energy-based learning objective to avoid these limitations. Finally, we demonstrate the details about network architecture to implement our method.

### 3.1 PRELIMINARIES

**Problem Formulation.** For a task $\mathcal{T}$, there are $N$ expert demonstrations $\{\pi_i\}_{i=1}^N$. Each demonstration $\pi_i$ consists of a sequence of state-action pairs $\{o_t, a_t\}_{t=1}^T$, where $a_t$ denotes the action, $o_t$ represents the observation, $T$ is the action sequence length. We formulate robot policy learning as an action sequence prediction problem. The aim is to train a model to minimize the error in future actions conditioned on historical states. Specifically, policy learning minimize the imitation learning loss $\mathcal{L}_{im}$ formulated as

$$\mathcal{L}_{im} = \mathbb{E}_{\pi_i \sim \mathcal{T}} \left[ \sum_{t=0}^T \mathcal{L} \left( f_\theta(a_{t:t+H-1}|o_{t'<t}), a_{t:t+H-1} \right) \right] \tag{1}$$

where $H$ is the prediction horizon, $t$ and $t'$ denote the current and previous time step, respectively. $\mathcal{L}$ represents a supervised action prediction loss, and $\theta$ represents the learnable parameters of the policy network $f_\theta$. Based on above problem formulation, the existing works (see Figure 2) mainly differ in how the $\mathcal{L}$ and $H$ are defiend, discussed as follows.

**Autoressive Policy.** By default, autoregressive-based policies predict action sequences in an autoregressive manner, with $H$ set to 1. Additionally, due to the discrete nature of action tokens, cross-entropy loss is typically used as the default objective $\mathcal{L}$. However, using discrete action tokens often sacrifices fine-grained action details, making it challenging for robotic tasks that require high-precision control.

**L1/L2 Policy.** To circumvent the use of discrete action tokens, L1/L2 policies have been proposed. By using L1/L2 loss as the objective $\mathcal{L}$, these methods can predict continuous actions. This makes the learned policy well-suited for high-precision manipulation tasks. However, it struggles with multimodal action distributions due to its unimodal modeling characteristics.

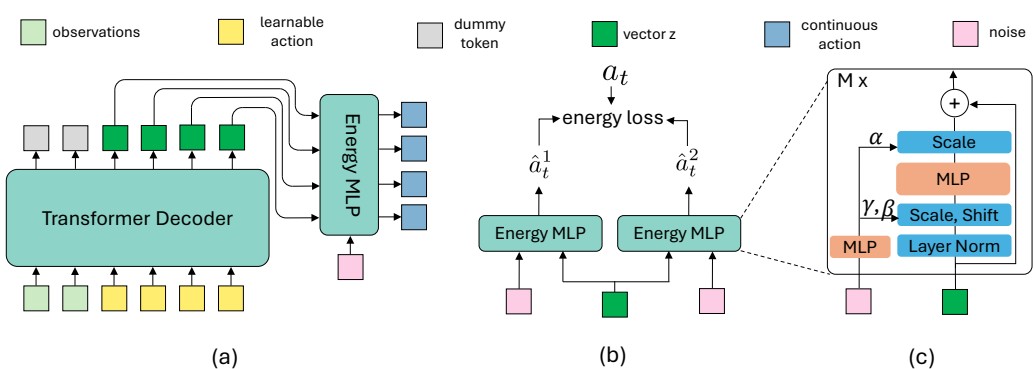

Figure 3: **Overview of Energy Policy.** (a) The architecture of the energy policy primarily consists of a transformer decoder and an energy MLP. The transformer decoder takes observations and learnable action tokens as input, producing a sequence of vectors $\{z_t\}_{t=1}^{H}$. The energy MLP then predicts the action sequence $\{\hat{a}_t\}_{t=1}^{H}$ by taking $\{z_t\}_{t=1}^{H}$ as input, conditioned on noise samples. (b) The energy loss is computed based on two sampled actions, $\hat{a}_t^1$ and $\hat{a}_t^2$, along with the ground-truth action $a_t$. These two action samples are generated from the same $z_t$ using different noise inputs. (c) The energy MLP is composed of several residual blocks, each incorporating adaLN for noise injection and modulation.

**Diffusion Policy.** In diffusion-based policies, action prediction is modeled as a denoising process, which makes it easier to handle multimodal action distributions. It uses denoising loss as the objective $\mathcal{L}$ and predicts a sequence of actions with $H > 1$ simultaneously. However, diffusion-based policies suffer from the need for multiple denoising steps, making them less suitable for high-frequency robotic tasks.

### 3.2 ENERGY POLICY

To address the limitations of existing approaches, we propose an energy-based policy that uses energy scores as the learning objective. With energy scores, the model can learn multimodal action distributions during training. During inference, continuous actions can be sampled in a single forward pass. Additionally, we introduce a decoder-only transformer specifically designed for energy policy.

#### 3.2.1 ENERGY LOSS

Consider two probability distributions $p$ and $q$ in $\mathbb{R}^d$. A scoring rule is a function $S(p, y)$ that assigns a score to the distribution $p$ based on the observed data $y \sim q$ (Gneiting & Raftery, 2007a). The expected score $S(p, q)$ is defined as $S(p, q) := \mathbb{E}_{y \sim q}[S(p, y)]$. A scoring rule is called strictly proper if the expected score is minimized if and only if $p = q$.

A commonly used class of strictly proper scoring rules is the energy scores (Székely, 2003), defined as

$$S(p, y) = -\mathbb{E}[\|x_1 - x_2\|^\alpha] + 2\mathbb{E}[\|x - y\|^\alpha] \tag{2}$$

where $x_1, x_2, x \in \mathbb{R}^d$ are independent samples drawn from the model distribution $p$, and $\|\cdot\|$ denotes the Euclidean norm. By allowing for an index $\alpha \in (0, 2)$, $S(p, q)$ is minimized if and only if $p = q$, which implies that the model distribution is consistent with the data distribution.

For action prediction, the model takes an observation as input and outputs a prediction distribution $p$ for the action $a_t$. To obtain an unbiased estimate of the energy score $S(p, q)$, two independent samples, $\hat{a}_t^1$ and $\hat{a}_t^2$, are drawn from the model distribution $p$. The energy loss for action prediction is then defined as:

$$\mathcal{L}(p, a_t) = \|\hat{a}_t^1 - a_t\|^\alpha + \|\hat{a}_t^2 - a_t\|^\alpha - \|\hat{a}_t^1 - \hat{a}_t^2\|^\alpha \tag{3}$$

This loss objective incentivizes the model to generate samples close to the target action, while maintaining the diversity between independent samples. To implement the energy loss, we introduce an energy-based MLP within the transformer architecture.

### 3.2.2 Transformer with Energy MLP

As shown in Figure 3, our model takes observations and learnable action tokens as input. These input tokens are also combined with learnable position tokens. The decoder-only transformer then generates a sequence of vectors $\{z_t\}_{t=1}^H$, each corresponding to a learnable action token. For each vector $z_t$, the corresponding continuous ground-truth action is denoted as $a_t$.

Given ground-truth action $a_t$, we introduce a dedicated energy MLP specifically designed for the energy loss. This network takes $z_t$ as input, together with two random noise samples drawn from a uniform distribution, and outputs two candidate actions $\hat{a}_t^1$ and $\hat{a}_t^2$. The energy MLP consists of several residual blocks (He et al., 2016). Each block sequentially applies LayerNorm (LN) (Ba et al., 2016), a linear layer, SiLU (Elfwing et al., 2018), another linear layer, and a residual connection. To inject stochasticity, we adopt adaLN-Zero blocks (Peebles & Xie, 2022), which condition on noise inputs and perturb the predictions $z_t$ through shift, scale, and gate operations. This design enables effective conditioning on noise, which not only introduces richer stochasticity but also enhances the model's expressive power.

At inference time, this energy MLP takes $z_t$ and a single noise sample to directly predict the corresponding continuous action, ensuring consistency between training and deployment while avoiding iterative refinement.

## 4 Evaluation

We conduct a comprehensive evaluation of our method across a diverse set of robotic tasks in both simulated and real-world environments. These tasks include both single-task and multi-task settings, utilizing image-based and state-based observations. We compare our approach against several state-of-the-art baselines, including both diffusion-based and auto-regressive methods. The evaluation considers key metrics such as success rate, inference time, and model size. Our experimental study aims to address the following research questions:

- How does our method compare to state-of-the-art approaches in terms of inference speed and task success rate?

- How well does our method perform robotic tasks in real-world environments?

- Can our energy loss effectively learn the multimodal action distribution?

### 4.1 Simulation Environment

We evaluate our method across diverse settings, including single-object manipulation, long-horizon planning, high-precision control, and dual-arm coordination, using Robomimic (Mandlekar et al., 2021), Franka Kitchen (Gupta et al., 2019), MimicGen (Mandlekar et al., 2023), and PushT (Chi et al., 2023). Unless otherwise specified, the models are trained for 400 epochs with a batch size of 1024 and $\alpha = 1.0$, using an energy MLP head with a depth of 3 and a width of 512. At inference, the model predicts an action sequence of length 16, from which the first 8 actions are executed. Baseline models are trained and evaluated following their original protocols. Performance is reported as the average success rate over the best three checkpoints. Inference efficiency is measured on an NVIDIA RTX 4090 GPU by averaging the runtime for generating 8 executable actions, with each test repeated three times.

#### 4.1.1 Single Object Manipulation on Robomimc

**Setup.** Robomimic offers a diverse set of tasks; in this section, we focus on three representative single-object manipulation tasks: Lift, Can, and Square. For each task, we use 200 expert demonstrations collected via teleoperation in simulation and evaluate performance under two observation modalities: image-based (RGB inputs from eye-in-hand and third-person views) and state-based (low-dimensional privileged information). We compare against three baselines: Implicit Behavior Cloning (IBC) (Florence et al., 2021), Diffusion Policy (DP-C, DP-T) (Chi et al., 2023), and CARP (Gong et al., 2024).

Table 1: Comparison of policy performance and efficiency across state-based and image-based Robomimic (Mandlekar et al., 2021) tasks.

| Policy | State-based | | | | | Image-based | | | | |
|---|---|---|---|---|---|---|---|---|---|---|
| | Lift-ph | Can-ph | Square-ph | Params(M) | Speed(s) | Lift-ph | Can-ph | Square-ph | Params(M) | Speed(s) |
| IBC (Florence et al., 2021) | 0.79 | 0.00 | 0.00 | 3.20 | 0.03 | 0.94 | 0.08 | 0.03 | 3.44 | 0.10 |
| DP-C (Chi et al., 2023) | **1.00** | 0.94 | 0.94 | 65.88 | 0.70 | **1.00** | 0.97 | 0.92 | 255.61 | 0.72 |
| DP-T (Chi et al., 2023) | **1.00** | **1.00** | 0.88 | 8.97 | 0.64 | **1.00** | **0.98** | 0.86 | 9.01 | 0.67 |
| CARP (Gong et al., 2024) | **1.00** | **1.00** | **0.98** | 0.65 | 0.07 | **1.00** | **0.98** | 0.88 | 7.58 | 0.11 |
| Ours | **1.00** | **1.00** | 0.97 | 0.73 | **0.01** | **1.00** | **0.98** | **0.95** | 11.51 | **0.03** |

Table 2: Performance and efficiency of different policies on Franka-Kitchen (Gupta et al., 2019) tasks.

| Policy | p1 | p2 | p3 | p4 | Params(M) | Speed(s) |
|---|---|---|---|---|---|---|
| IBC (Florence et al., 2021) | 0.99 | 0.87 | 0.61 | 0.24 | 3.28 | 0.05 |
| DP-C (Chi et al., 2023) | **1.00** | **1.00** | **1.00** | 0.96 | 66.94 | 0.91 |
| DP-T (Chi et al., 2023) | **1.00** | 0.99 | 0.98 | 0.96 | 80.42 | 0.84 |
| CARP (Gong et al., 2024) | **1.00** | **1.00** | 0.98 | **0.98** | 3.88 | 0.08 |
| Ours | **1.00** | **1.00** | **1.00** | 0.96 | 5.06 | **0.02** |

**Result.** As shown in Table 1, our model matches or surpasses the baselines in both state-based and image-based settings, while also delivering substantially faster inference. Specifically, our approach is $3.7\times \sim 7.0\times$ faster than the autoregressive baseline (CARP) and $22.3\times \sim 70.0\times$ faster than the diffusion-based baselines (DP-C and DP-T). Although achieving a low latency, IBC exhibits near-zero success rates on several tasks. In contrast, our method maintains high success rates comparable to the strongest baselines while eliminating the costly iterative sampling steps of diffusion. This demonstrates both efficiency and robustness across single-object manipulation tasks.

### 4.1.2 LONG HORIZON PLANNING ON FRANKA KITCHEN

**Setup.** To assess long-horizon, multi-task learning, we evaluate on the Franka Kitchen environment, which involves interaction with seven objects and 566 human demonstrations, each completing four tasks in arbitrary order. Only state-based inputs are used. We compare against three baselines: Implicit Behavior Cloning (IBC) (Florence et al., 2021), Diffusion Policy (DP-C, DP-T) (Chi et al., 2023), and CARP (Gong et al., 2024).

**Result.** As shown in Table 2, our model attains success rates that are comparable to or exceed those of the baselines across all tasks, while also maintaining the fastest inference speed. Specifically, our approach runs $4\times$ faster than the autoregressive baseline (CARP) and $42\times \sim 45\times$ faster than the diffusion-based baselines (DP-C and DP-T). These results highlight the efficiency and scalability of our design in complex, long-horizon environments.

### 4.1.3 MULTI-TASK LEARNING ON MIMICGEN

**Setup.** We evaluate our model on MimicGen (Mandlekar et al., 2023), a large-scale imitation learning benchmark, which extends Robomimic (Mandlekar et al., 2021) by providing 1K–10K demonstrations per task and broader initial state distributions. MimicGen comprises 12 MuJoCo (Todorov et al., 2012)-based robosuite tasks and 4 high-precision tasks from Isaac Gym Factory (Makoviychuk et al., 2021). Following prior work (Wang et al., 2024a; Gong et al., 2024), we evaluate on 8 robosuite tasks, each with 1K demonstrations. The baselines include two diffusion-based multitask models—Task-Conditioned Diffusion (TCD) (Liang et al., 2023) and Sparse Diffusion Policy (SDP) (Wang et al., 2024a), the latter employing a Transformer with a Mixture of Experts (MoE) (Shazeer et al., 2017)—as well as the autoregressive multitask variant of CARP (Gong et al., 2024). All models are conditioned on task labels to enable multitask learning.

**Result.** As shown in Table 3, our method achieves substantial improvements over the diffusion baseline, with about a 10% higher success rate and at least a $16.6\times$ faster inference. Moreover, it matches the success rate of the autoregressive baseline while delivering a $2.3\times$ faster inference. These results further support our first research question.

Table 3: Performance and efficiency comparison of different policies across 8 manipulation tasks in MimicGen (Chi et al., 2024).

| Policy | Coffee | Hammer | Mug | Nut | Square | Stack | Stack 3 | Threading | Avg. | Params (M) | Speed (s) |
|---|---|---|---|---|---|---|---|---|---|---|---|
| TCD (Liang et al., 2023) | 0.77 | 0.92 | 0.53 | 0.44 | 0.63 | 0.95 | 0.62 | 0.56 | 0.68 | 156.11 | 1.33 |
| SDP (Wang et al., 2024a) | 0.82 | **1.00** | 0.62 | 0.54 | 0.82 | 0.96 | 0.80 | 0.70 | 0.78 | 159.85 | 1.53 |
| CARP (Gong et al., 2024) | 0.86 | 0.98 | **0.74** | 0.78 | **0.90** | **1.00** | **0.82** | 0.70 | 0.85 | 16.08 | 0.18 |
| Ours | **0.89** | 0.98 | 0.69 | **0.86** | 0.89 | 0.99 | 0.75 | **0.79** | **0.86** | 17.85 | **0.08** |

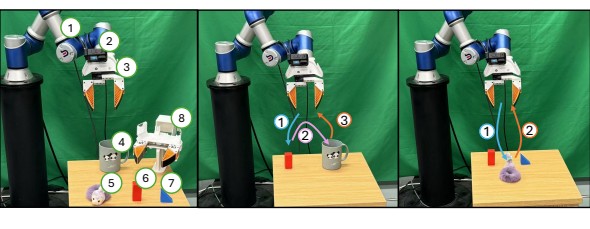
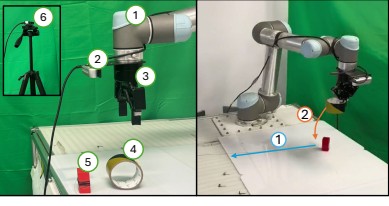

(a)                                          (b)

Figure 4: **Real-World Experiment Setup.** (a) Left: equipment and objects used in the static tasks, with each item labeled by a circled number in the upper-right corner. Middle: desired manipulation behavior for Task *Rabbit*. Right: desired manipulation behavior for Task *Cup* . (b) Left: equipment and objects used in the dynamic task, with each item labeled by a circled number in the upper-right corner. Right: desired manipulation behavior for Task *Catch*.

### 4.1.4 COMPARISON WITH OTHER EFFICIENT POLICIES

**Setup.** To further evaluate the performance of our method, we compare it against recent single-step robotic policies on more challenging tasks, including PushT, Square-mh, Square-ph, ToolHang-ph, Transport-mh, and Transport-ph, which challenge the model's ability to learn dual-arm and high-precision manipulation. The baselines include One-step Diffusion Policy (OneDP) (Wang et al., 2024c), IMLE Policy (IMLE) (Rana et al., 2025), Consistency Policy (CP) (Prasad et al., 2024), and VQ-BeT (Lee et al., 2024). These methods are designed to improve the efficiency of robotic policy inference by eliminating or reducing reliance on multi-step sampling procedures.

**Result.** As summarized in Table 4, our Energy Policy consistently outperforms all baselines in average success rate across diverse tasks. It achieves an average success rate of 0.87, clearly surpassing OneDP, IMLE, CP, and VQ-BeT, while maintaining competitive inference speed. Although VQ-BeT attains the lowest latency, this comes at the cost of a substantial drop in task performance, underscoring the advantage of our approach in balancing efficiency and effectiveness.

Table 4: Comparison of different efficient policies across tasks. **\*** For fair comparisons, we report the model parameter counts and inference speeds on the PushT task using an NVIDIA RTX 4090 GPU.

| Policy | PushT | Square-mh | Square-ph | Toolhang-ph | Transport-mh | Transport-ph | Avg. | Params(M)* | Speed(ms)* |
|---|---|---|---|---|---|---|---|---|---|
| Ours | **0.85** | 0.85 | **0.95** | **0.92** | **0.70** | **0.95** | **0.87** | 7.73 | 7.02 |
| OneDP-S (Wang et al., 2024c) | 0.82 | **0.86** | 0.93 | 0.85 | 0.69 | 0.91 | 0.84 | 251.51 | 9.33 |
| IMLE (Rana et al., 2025) | 0.59 | – | 0.82 | 0.81 | – | 0.90 | – | 75.75 | 7.63 |
| CP (Prasad et al., 2024) | 0.82 | – | 0.92 | 0.70 | – | – | – | 255.18 | 15.23 |
| VQ-BeT (Lee et al., 2024) | 0.68 | – | – | – | – | – | – | 4.37 | **4.09** |

### 4.2 REAL-WORLD ENVIRONMENTS

We evaluate our model on two robotic platforms to assess real-world manipulation performance. To systematically test different capabilities, we design three tasks: two static tasks (**Cup** and **Rabbit**) that assess precision and target selection under distraction, and one dynamic task (**Catch**) that evaluates real-time interaction. As the baseline, we use the CNN-based Diffusion Policy (Chi et al., 2023). Each model is evaluated from a single checkpoint, with success measured as the number of successful trials out of 20 per task. For speed evaluation, both models are deployed on the same machine with

Table 5: Comparison of policy performance and efficiency across real-world tasks.

| Policy | Static Tasks | | | | Policy | Dynamic Task | | |
| | Cup | Rabbit | Params(M) | Speed(s) | | Catch | Params(M) | Speed(s) |
| --- | --- | --- | --- | --- | --- | --- | --- | --- |
| DP-UMI (Chi et al., 2024) | 16/20 | 19/20 | 85.47 | 0.34 | DP-C (Chi et al., 2023) | 8/20 | 64.87 | 0.10 |
| Ours | 17/20 | 20/20 | 68.65 | 0.10 | Ours | 13/20 | 11.50 | 0.02 |

an NVIDIA RTX 1080Ti GPU, and we report the average inference time for generating 8 executable action steps.

### 4.2.1 STATIC TASKS

**Setup.** As shown in Figure 4 (a, left), an Emergen CR3 robotic arm[1] is equipped with a 3D-printed Universal Manipulation Interface (UMI) gripper[3] (Chi et al., 2024) and a wrist-mounted GoPro Hero 9 camera[2] with a fisheye lens. we collect 150 human demonstration trajectories per task using the UMI gripper[8]. As illustrated in Figure 4 (a, center and right), we design the following real-world tasks: **Cup.** The workspace contains a green cup[4] and a red block[6] on a tabletop. The robot must grasp the block and place it inside the cup. While the cup remains fixed, the block's initial position is perturbed to introduce variability. This task evaluates precise pick-and-place under positional uncertainty. **Rabbit.** The workspace includes a soft rabbit toy[5], a red rectangular block[6], and a blue triangular block[7], which are randomly assigned to three predefined locations with small perturbations. The robot must identify and grasp the rabbit among distractors, testing the robustness to spatial variation and target ambiguity. As the baseline, we use the CNN-based Diffusion Policy (DP-UMI) (Chi et al., 2024).

**Results.** As shown in Table 5, our model achieves a slightly higher success rate than the baseline, while maintaining a $3.4\times$ inference speedup. Although the perturbations introduced during evaluation are relatively small, DP-UMI (Chi et al., 2024) occasionally fails, primarily due to slight gripping inaccuracies when the state deviates from the training distribution. In contrast, Energy Policy executes more reliably under the same conditions, underscoring its robustness in real-world robotic tasks.

### 4.2.2 DYNAMIC TASK

**Setup.** As shown in Figure 4 (b, left), a UR5 robotic arm[1] is equipped with a Robotiq 2F-85 adaptive gripper[3] and a wrist-mounted Intel RealSense D435i camera[2]. An additional RealSense D435i camera[6] mounted on a fixed stand provides a third-person view. We collect 100 teleoperated demonstrations using a 3D SpaceMouse. As illustrated in Figure 4 (b, right), we design the following task: **Catch.** The workspace contains a red block[5] attached to a string and a ring[4] held by the gripper. A human drags the block across the table at random speeds from right to left, and the robot must intercept it with the ring before it exits the camera's field of view. This task challenges the policy's inference speed and its ability to interact with fast-moving objects in real time. As the baseline, we use the CNN-based Diffusion Policy (DP-C) (Chi et al., 2023).

**Results.** As shown in Table 5, our model outperforms the CNN-based Diffusion Policy on the dynamic **Catch** task (13/20 vs. 8/20) with a $5\times$ faster inference speed (0.02s vs. 0.10s). The performance gap is more pronounced than in static tasks, since multi-step diffusion models struggle with latency. In contrast, our low-latency predictions enable reliable performance under time-sensitive conditions.

### 4.3 MODELING MULTI-MODAL BEHAVIOR

To evaluate the model's ability to capture multimodal behavior, we design an initial state in the PushT environment where the task can be successfully completed by moving either left or right. We then sample 50 trajectories from the Energy Policy to examine its ability to model multi-modal behavior. The visualizations are shown in Figure 5, Energy Policy generates smooth trajectories covering both modes.

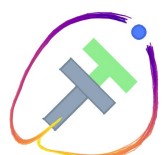

Figure 5: Rollout of the first 40 steps of the samples.

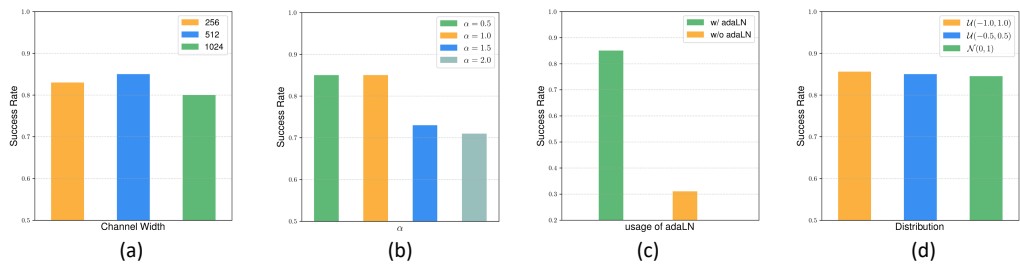

Figure 6: **Ablation Study Results.** (a) Success rate as a function of MLP channel width. (b) Success rate as a function of $\alpha$. (c) Success rate with and without AdaLN. (d) Success rate under different noise injection distributions.

## 4.4 ABLATION STUDY

We conduct an ablation study to evaluate the contributions of the newly introduced components: the *Energy MLP* and the *Energy Loss*. All experiments are conducted on the *Square-mh* task from Robomimic (Mandlekar et al., 2021). Models are trained for 400 epochs with a batch size of 1024. For experiments involving the energy loss, we set the weighting coefficient $\alpha = 1.0$. Performance is assessed by averaging the success rates of the top three checkpoints. We investigate: (i) the impact of the Energy MLP channel width, (ii) the effect of the coefficient $\alpha$, (iii) the role of adaLN, and (iv) the effect of different noise injection distributions.

**Channel Size of Energy MLP.** We investigate the impact of the Energy MLP channel width on model performance. Starting with a width equal to the token embedding size (256), we progressively increase the width to 512 and 1024. Performance improves from 0.83 to 0.85 when increasing the width to 512, but further increasing it to 1024 results in a decline to 0.80. We hypothesize that this degradation is due to overfitting. Based on these results, we adopt 512 as the default MLP width for all evaluations in the image-based simulation environment.

**Choice of the coefficient $\alpha$.** In our main experiments, we set $\alpha$ to 1.0 empirically. To assess the sensitivity to this hyperparameter, we conduct an ablation study by varying its value. As shown in the Square-mh task, performance remains stable when reducing $\alpha$ from 1.0 to 0.5 (success rate 0.85), but degrades when increasing it to 1.5 (0.73) or 2.0 (0.71). Thus, we adopt $\alpha = 1.0$ as the default setting in all experiments.

**Role of Adaptive Layer Normalization (adaLN).** To validate its effectiveness, we compare our approach with a baseline that simply concatenates Gaussian noise with the output vector from the Transformer decoder, instead of applying adaptive layer normalization. We evaluate the success rate on the Square-mh task. Removing adaLN leads to a drastic performance drop from 0.85 to 0.31, highlighting that adaLN is crucial for effectively modeling noise conditioning.

**Choice of Noise Distribution.** By default, the noise injected into the Energy MLP is sampled from a uniform distribution in the range $[-0.5, 0.5]$. We also evaluated two alternative noise sources: a uniform distribution over $[-1.0, 1.0]$ and a Gaussian distribution $\mathcal{N}(0, 1)$. In all cases, the resulting success rates differ by less than 1%, indicating that our method is robust to variations in the noise injection distribution.

## 5 CONCLUSION

Energy Policy offers a novel approach to multimodal action modeling, making it well-suited for robotic tasks that demand high-precision manipulation. Moreover, thanks to its underlying mechanism, it can generate a sequence of continuous actions in a single forward pass, resulting in significantly faster inference. Our strong performance across a variety of robotic tasks demonstrates both the effectiveness and efficiency of the proposed method.

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

Table 6: Evaluation on Meta-World.

| Policy | Disassemble | Pick Place Wall | Shelf Place | Stick Pull | Stick Push | Avg. | Speed (ms) |
|---|---|---|---|---|---|---|---|
| Energy Policy (Ours) | 0.95 | 0.85 | 0.30 | 0.70 | 0.75 | 0.71 | 7.72 |
| DP3 | 0.95 | 0.90 | 0.25 | 0.60 | 0.80 | 0.70 | 73.33 |

# A  APPENDIX

## A.1  MANIPULATION TASKS ON META-WORLD

**Setup.**   We further extend our evaluation by adapting our method to the 3D Diffusion Policy Ze et al. (2024b) framework and testing on the Meta-World benchmark Yu et al. (2019). Following prior work Ze et al. (2024b), we evaluate five challenging tasks: *Disassemble*, *Pick Place Wall*, *Shelf Place*, *Stick Pull*, and *Stick Push*. This setting validates the ability of our model to handle complex, high-dimensional observation spaces.

**Implementation.**   We adopt the same observation setup, encoder, and network architecture as 3D Diffusion Policy (DP3), with one key modification: the diffusion timestep embedding is removed, and a two-layer energy MLP (width 256) is added as the output head. Both models are trained for 1000 epochs with a batch size of 256.

**Results.**   Table 6 summarizes the results. Our method achieves a higher average success rate than DP3 (0.71 vs. 0.70), while reducing inference latency by more than an order of magnitude (7.72 ms vs. 73.33 ms). These results highlight that our approach maintains competitive performance while delivering substantial efficiency gains in high-dimensional observation spaces settings.

## A.2  EXPERIMENTS IMPLEMENTATION DETAILS

### A.2.1  SINGLE-TASK SIMULATION (ROBOMIMIC, FRANKA KITCHEN, PUSHT)

We adopt the observation space configuration from Diffusion Policy (Chi et al., 2023). For image-based tasks, we use an observation horizon of 2, consisting of multiview RGB images and proprioceptive states. For state-based tasks, the observation horizon is 2 (Robomimic (Mandlekar et al., 2021)) or 4 (Franka Kitchen (Gupta et al., 2019)), using low-dimensional object state vectors as input. The action prediction horizon is set to 16, with only the first 8 actions executed during the evaluation. Our model, a decoder-only transformer with an energy-based MLP head, is trained with a batch size of 1024 for 400 epochs (image-based) or 600 epochs (state-based) using $\alpha = 1.0$. For the decoder-only Transformer, we use the same depth and token embedding dimension as DP-T. Baseline models are trained and evaluated following the original protocols.

### A.2.2  MULTI-TASK SIMULATION (MIMICGEN)

We follow the observation space setup from Sparse Diffusion Policy (Wang et al., 2024a), using an observation horizon of 2 for image and robot pose input. Our model employs the same architecture as in the single-task evaluation, augmented with a task-class token and doubled network depth. Training is performed for 400 epochs with a batch size of 1024 and $\alpha = 1.0$. Baseline models are trained and evaluated according to their respective protocols. As in the single-task setting, the model predicts an action sequence of length 16, from which the first 8 actions are executed.

### A.2.3  REAL-WORLD EXPERIMENTS

For all real-world tasks, we use an observation horizon of 2, incorporating camera RGB images and proprioceptive states.

For the static tasks (*Cup* and *Rabbit*), our model retains the simulation architecture of a single task with modifications to address real-world complexity and noise: the transformer embedding size is tripled and the MLP channel width is doubled. Training is carried out for 150 epochs with a batch size of 64 and $\alpha = 1.0$. The baseline model follows its default architecture and training configuration. Both models predict an action sequence of length 16, with the first 8 actions executed.

For the dynamic task (*Catch*), our model uses the same architecture as in the single-task simulation setting without modification to the Transformer embedding size or MLP channel width. Training is carried out for 200 epochs with a batch size of 64 and $\alpha = 1.0$. The baseline model follows the configuration specified in its original implementation for real-world PushT.

### A.3 EXECUTION VISUALIZATION

We provide qualitative visualizations of successful rollouts across all benchmarks. In each figure, a row corresponds to a task rollout, with the leftmost frame showing the initial state and the rightmost frame showing the final state.

### A.3.1 ROBOMIMIC

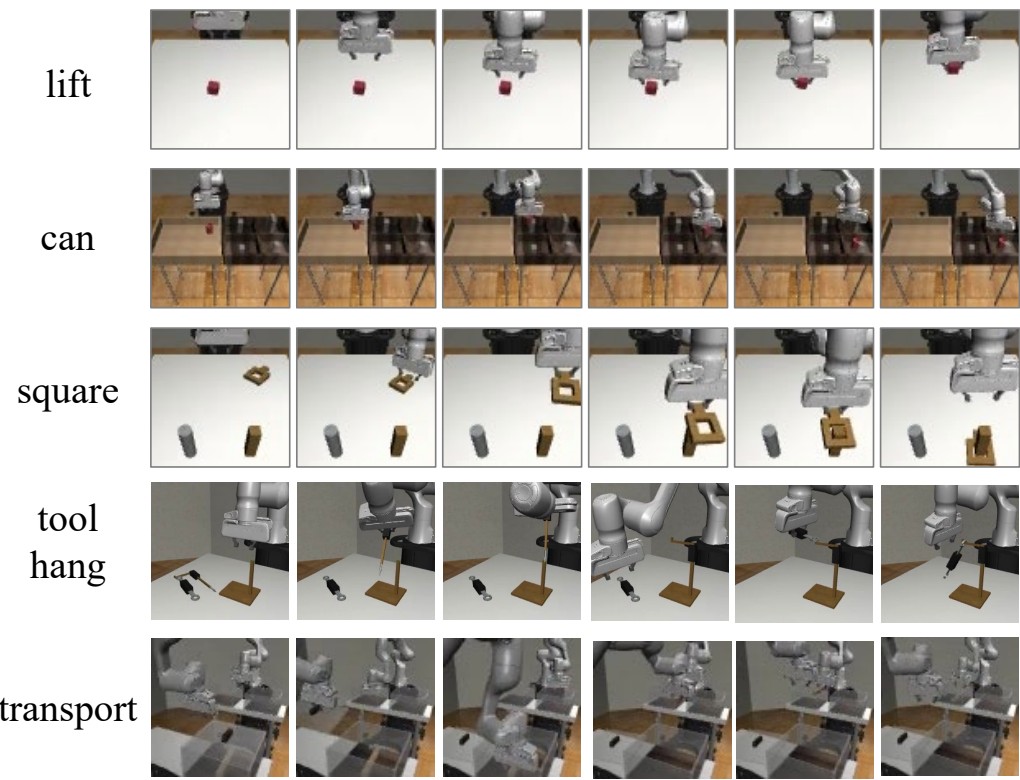

Figure 7: Visualization of Robomimic experiments.

## A.3.2 FRANKA KITCHEN

open oven

move kettle

heat on 1

heat on 2

light on

open cabinet 1

open cabinet 2

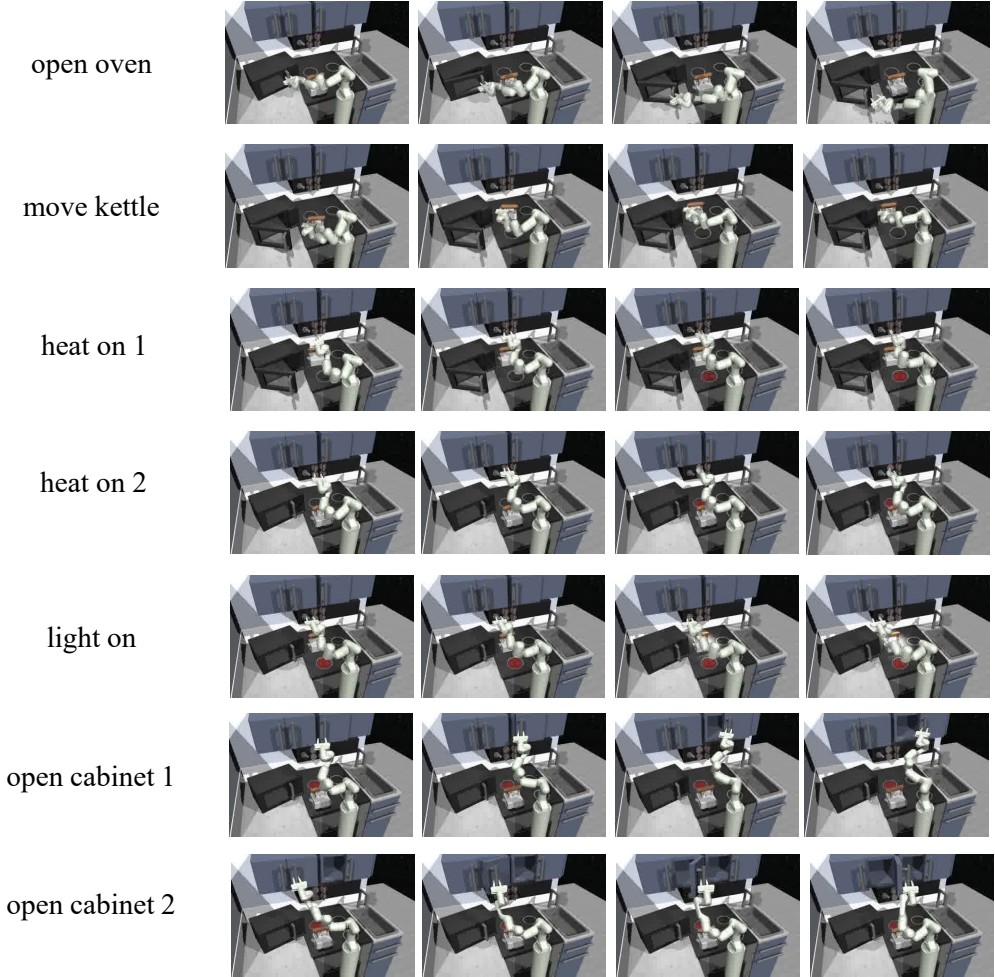

Figure 8: Visualization of Franka Kitchen subtasks.

### A.3.3 MIMICGEN

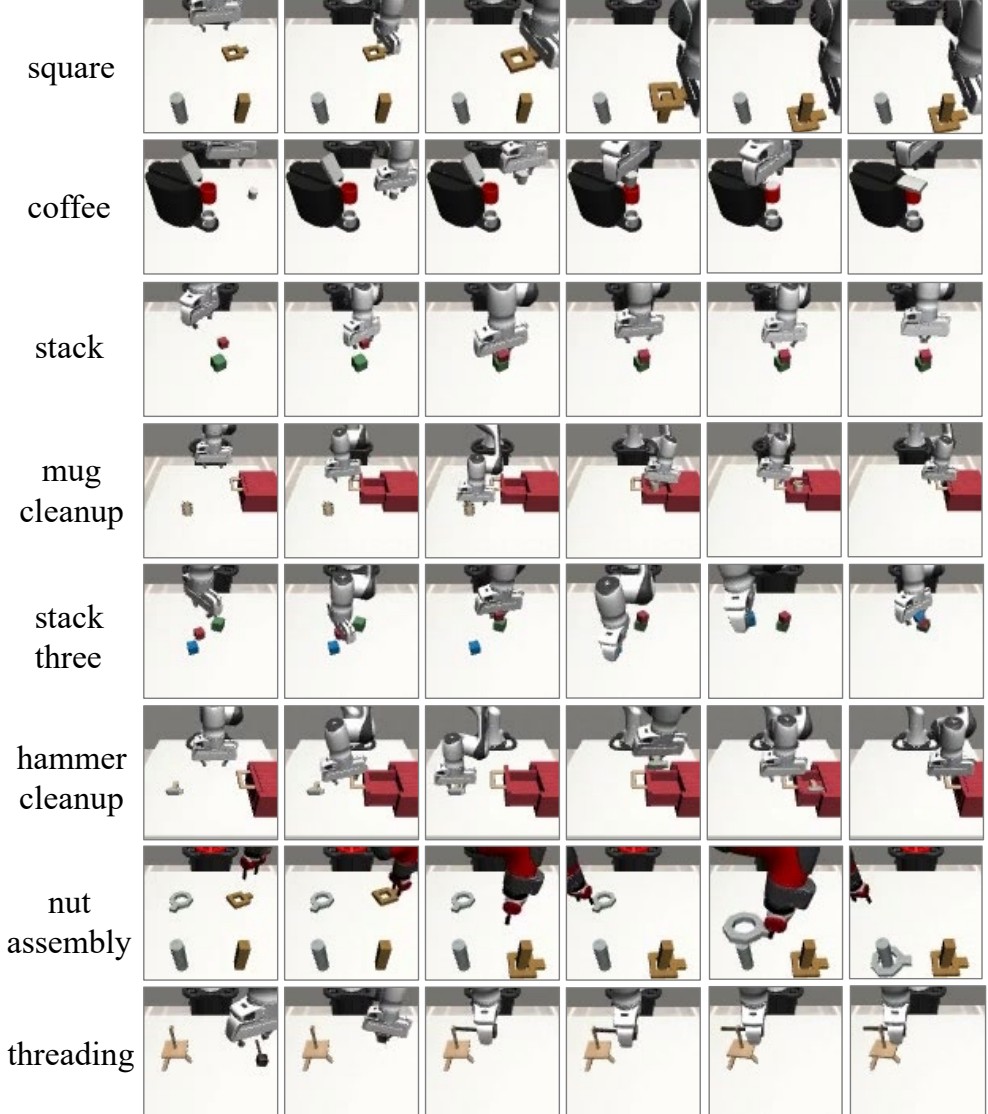

square

coffee

stack

mug
cleanup

stack
three

hammer
cleanup

nut
assembly

threading

Figure 9: Visualization of MimicGen multi-task experiments.

### A.3.4 META-WORLD

disassemble

pick-place-wall

shelf-place

stick-pull

stick-push

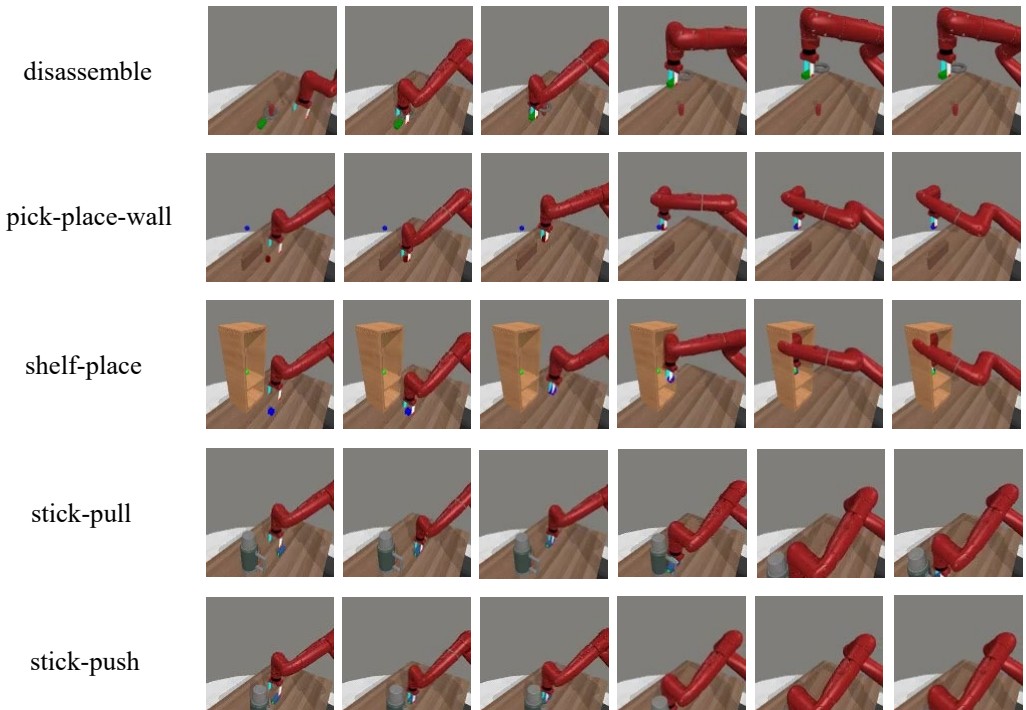

Figure 10: Visualization of Meta-World experiments.

### A.3.5 REAL WORLD (STATIC)

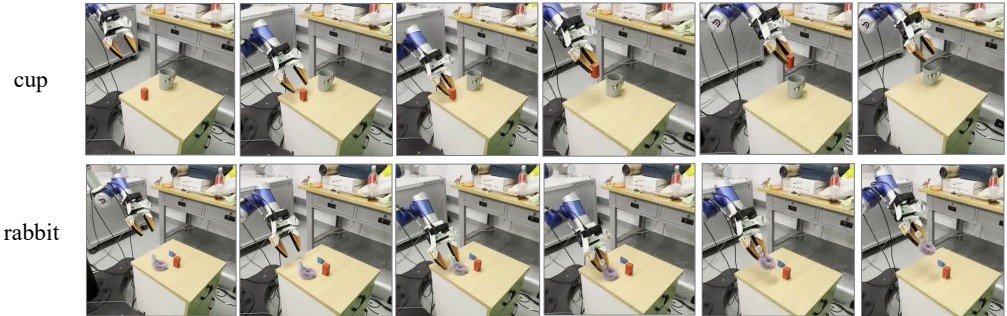

Figure 11: Visualization of real-world static task experiments.

### A.3.6 REAL WORLD (DYNAMIC)

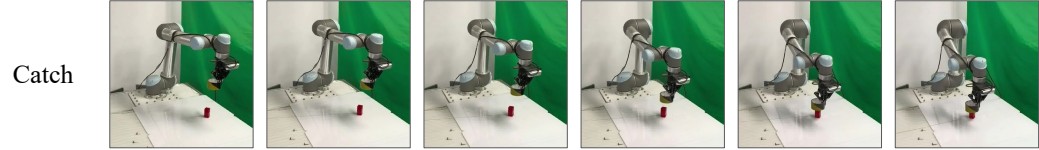

Figure 12: Visualization of real-world dynamic task experiments.

## B LLM USAGE

Large Language Models (LLMs) were used for grammar checking and writing refinement.

