# OpenReview forum: "Fast Visuomotor Policy for Robotic Manipulation"
_ICLR.cc/2026/Conference — Submitted to ICLR 2026_

### Official Review · Reviewer_yAJK · 2025-10-29

**Soundness:** 4
**Presentation:** 3
**Contribution:** 4
**Rating:** 6
**Confidence:** 3

**Summary:**

The paper proposes using energy score-based policies as a way to obtain multi-modal continuous actions while keeping a reasonable inference speed. The main motivation is to bypass the need for lengthy diffusion online or the loss of accuracy via distillation or loss of granularity and accuracy based on discretization. The proposed approach is compared to multiple baselines in single and mutli-task settings in simulation as well as 3 real-world tasks.

**Strengths:**

- The paper is very well written and is an enjoyable read
- Finding methods that appropriately tradeoff inference speed and performance is very important for policy learning, specially in robotic manipulation
- The proposed approach is very simple making it an elegant solution to the problem
- The results are quite promising, almost consistently showing a mostly equivalent (sometimes worse, sometimes better) performance in comparison to the baselines, while improving on inference speed
- Experiments include ablations of the introduced hyperparameters. This makes building on top of this work by properly understanding a lot better.

**Weaknesses:**

- Novelty is limited
- Table 4 is missing some values without an explanation for why this is the case
- The papers lacks a deeper analysis of the experimental results, for instance in some cases the proposed approach outperforms the baselines in others it doesn't. It would be interesting if the authors can attempt to provide an understanding of which settings are most and least suited for their method

**Questions:**

Can you provide a deeper discussion based on your results, on what settings most favor your method, and which settings can be leading to drop in performance? to rephrase, do you have an intuition on what to blame the drop in performance in certain tasks on?

---

> ### Author Response · Authors · 2025-11-23
> **Response to Reviewer yAJK**
>
> **1. Novelty is limited**
>
> Thank you for the comment. We clarify the novelty of our contribution.
>
> **A new learning formulation not covered by existing policy families.** Prior visuomotor policies are typically based on (1)  L1/L2 regression (uni-modal), (2) autoregressive token models (slow, discrete), (3) diffusion/flow-matching methods (multi-step sampling, high latency). Our approach introduces a new formulation: single-step multimodal continuous action generation trained with a strictly proper probabilistic objective (energy score).
>
> **Simple but effective architectural realization.** We instantiate this formulation using a noise-conditioned action head combined with the energy-score objective, enabling the policy to map a low-dimensional noise vector to diverse action hypotheses within a single forward pass. Although architecturally simple, this design is essential for unlocking multimodal behavior without the sequential overhead of diffusion or autoregressive policies.
>
> **2. Table 4 is missing some values without an explanation for why this is the case**
>
> Thank you for the comment. We clarify the missing entries in Table 4.
>
> The dashes correspond to results that **were not reported** in the original papers for those methods. Several accelerated single-step policies (e.g., IMLE Policy, Consistency Policy, VQ-BeT) only provide numbers for a subset of tasks in their official publications. To ensure fairness and avoid introducing implementation-dependent variance, we follow their reported results exactly rather than attempting to reproduce missing entries. We will add a brief note in the table caption to make this explicit.
>
> **3. The papers lacks a deeper analysis of the experimental results, for instance in some cases the proposed approach outperforms the baselines in others it doesn't.**
>
> Thank you for the comment. We clarify the intended evaluation perspective.
>
> **(1) The primary goal of the paper is to achieve high-speed, single-step inference while maintaining comparable performance.**
>
> As stated in the title and introduction, the contribution is a **single-step, low-latency** alternative to multi-step generative policies. Under this speed–accuracy view, the results are consistent: Energy Policy offers **2.3×–6.5× faster inference** while delivering performance that is on par with the strongest baselines across diverse settings.
>
> **(2) Small performance fluctuations across tasks are expected and typical in efficiency-oriented policies.**
>
> Across Robomimic, Kitchen, MimicGen, PushT, and real-world tasks, the proposed method is sometimes slightly better and sometimes slightly worse than individual baselines. This variation is normal given the diversity of tasks and control regimes. Importantly, in all cases, our method preserves strong task success while offering a **substantial reduction in latency**, which is the key motivation of this work.
>
> In short, the experiments aim to show that Energy Policy achieves dramatically faster inference while maintaining competitive accuracy, which is the central contribution of the work.
>
> **4. Can you provide a deeper discussion based on your results, on what settings most favor your method, and which settings can be leading to drop in performance? to rephrase, do you have an intuition on what to blame the drop in performance in certain tasks on?**
>
> Thank you for the question. We summarize the strengths and limitations of our method.
>
> **(1) Strengths.**
>
> Energy Policy is strongest in latency-sensitive or high-frequency settings, including dynamic tasks, fast-reaction control, and fine-grained continuous motions. In these scenarios, single-step generation avoids the delay introduced by multi-step denoising, resulting in at least a 2.3× reduction in latency while maintaining comparable performance across benchmarks.
>
> **(2) Limitations.**
>
>  Performance drops primarily occur in scenarios that are challenging for all visuomotor policies, such as heavy occlusion, ambiguous observations, or very long-horizon multi-stage interactions. These stem from general task complexity rather than a specific limitation of our formulation.
>
> In short, our method excels when fast reaction is critical, and small variations arise mainly in universally difficult settings.

---

### Official Review · Reviewer_JKTV · 2025-10-30

**Soundness:** 2
**Presentation:** 2
**Contribution:** 3
**Rating:** 2
**Confidence:** 5

**Summary:**

In this work, the authors introduced a visuomotor manipulation policy named Energy Policy. Contrary to previous works like distilled diffusion policy or other autoregressive methods, Energy Policy naturally only requires one single forward pass to predict multi-modal actions. The whole framework is lightweight and simply implemented with an energy MLP and a transformer decoder, which are optimized by the energy score. Experiments are conducted in both simulation and real-world settings, through which Energy Policy exhibits its comparable performance and efficiency to some degree.

**Strengths:**

- The whole idea and implementation of the Energy Policy is simple but quite effective. Besides, it naturally requires only one single pass to predict actions without the need for distillation.
- Using the energy score as the learning objective, the model can not only generate samples close to the target action but also maintain a certain level of diversity between independent samples.
- Experiments are conducted in both simulation and real-world settings. Simulation experiments include RoboMimic, Franka Kitchen, MimicGen, and PushT. Real-world experiments also have more than one setting. The speed advantage of Energy Policy is demonstrated through these settings.

**Weaknesses:**

- Although experiments are conducted in various settings, the tasks that the authors choose are quite simple, and most of them are just some basic atomic skills. For example, in the tasks from RoboMimic, state-based methods have already achieved 1.0 success rates. Thus, they cannot truly reflect the proposed methods’ effectiveness. So are the cases for Franka Kitchen.
- The selection of baselines across different environments is not consistent, and many comparisons are not complete. For instance, why are the original DP’s performances not shown in Table 4? Besides, multiple positions are blank and filled with dashes.
- Similarly, the real-world experiments also have such kinds of problems. The so-called dynamic tasks are actually still static ones.
- The writing of this work requires improving. Some expressions are just too over, e.g., in line 60 “...making it computationally prohibitive…”

**Questions:**

- Improvements on single-task policies may not be enough at current timesteps. Since the architecture is quite like some VLA models, do you have any plans to apply the energy policy to them to replace the flow matching method during post-training?
- Could you please provide more experimental results involving more complex manipulation tasks during the rebuttal stage?
- Can you provide some insights regarding your selection of the coefficient $\alpha$ from the perspective of data distribution other than just experience?

---

> ### Author Response · Authors · 2025-11-23
> **Response to Reviewer JKTV**
>
> **1. the tasks that the authors choose are quite simple, and most of them are just some basic atomic skills.**
>
> Thank you for the comment. We clarify three important points.
>
> **(1) The benchmarks used are fully appropriate for evaluating our main contribution—
>  achieving substantially lower inference latency without degrading performance.**
>
> The purpose of the experiments is to test whether **a single-step, continuous multimodal policy** can achieve **faster inference** than diffusion, autoregressive, consistency-style, and accelerated single-step policies, **without loss in success rate**. Even when baseline success rates are high, they still provide a clean and fair setting for measuring this speed–performance trade-off, which is the central claim of the paper.
>
> **(2) Several benchmarks in fact require nontrivial multimodal and long-horizon behavior.**
>
> **MimicGen** introduces broad initial-state distributions and multi-task variability across 8 heterogeneous manipulation skills. **PushT, Square-mh, ToolHang, Transport** require fine-grained, multi-step, and often multimodal control patterns. **Real-world Catch** contains dynamic, high-speed interactions where small timing variations lead to divergent trajectories. **Meta-World (Appendix A)** further extends complexity with high-dimensional 3D visual observations and diverse multi-step manipulation tasks.
> These tasks extend well beyond “basic atomic skills” and require handling variability, multimodality, and high-frequency decision-making.
>
> **(3) Additional experiments further reinforce that the method scales beyond these benchmarks.**
>
>  To strengthen the empirical picture, we also integrate Energy Policy into a SmolVLA-style vision-language-action model, which operates under **high-dimensional visual inputs, broad behavioral variability, and long-horizon reasoning**. As shown in **Table R1**, Energy Policy still provides a **substantial inference-time reduction** compared to the original flow-matching head, while **maintaining higher performance**.
> This result further supports that our single-step formulation remains effective well beyond the “atomic skill” regime, demonstrating scalability to richer, more complex manipulation distributions.
>
> **Table R1: Inference speed and success rate comparison between SmolVLA and SmolVLA + Energy.**
> | Method           | Speed (s) | Spatial | Object | Goal | Long | Avg   |
> |------------------|-----------|---------|--------|------|------|-------|
> | SmolVLA          | 0.038     | 90      | 96     | 92   | 71   | 87.25 |
> | SmolVLA + Energy | **0.012**     | 95      | 95     | 94   | 77   | **90.25** |
>
> **2. The selection of baselines across different environments is not consistent.**
>
> Thank you for the comment. We clarify the baseline selection and missing entries below.
>
> **(1) Table 4 is intentionally restricted to single-step/accelerated policies.**
>
> Its purpose is to compare methods that share the same **one-pass inference paradigm** (OneDP, IMLE Policy, CP, VQ-BeT). Original diffusion models (DP-C/DP-T) are multi-step and therefore evaluated separately in Tables 1–3, which are dedicated to full diffusion and autoregressive baselines. We will clarify this separation in the revision.
>
> **(2) Dashes indicate results not reported in the original papers.**
>
> Several accelerated-policy methods only provide numbers for a subset of tasks. We follow the official reporting to avoid introducing potentially unfair comparisons—reproducing these methods on new tasks can be sensitive to undocumented implementation details and hyperparameters. We will make this explicit in the table caption.
>
> **(3) Each environment uses baselines appropriate for its evaluation goal.**
>
> Tables 1–3 focus on general visuomotor performance and therefore include state-of-the-art diffusion and autoregressive models. Table 4 targets efficient single-step or accelerated policies, so we compare only methods in that computational class. For real-world experiments, we adopt DP-C as the standard real-time diffusion baseline. The appendix further includes Meta-World to demonstrate scalability in a more complex setting.
>
> In summary, baseline selection is **consistent by evaluation objective**, and missing entries reflect **unreported numbers**, not omissions.

---

> > ### Author Response · Authors · 2025-11-23
> >
> > **3. Similarly, the real-world experiments also have such kinds of problems. The so-called dynamic tasks are actually still static ones.**
> >
> > Thank you for the comment. In recent visuomotor control work, a task is typically considered dynamic when the target state evolves during execution and action timing materially affects success. The “dynamic picking” task in BID [1] follows this definition: the robot must intercept a continuously moving object. Our Catch experiment is of the same nature—the target moves with varying velocities throughout the episode, and small latency differences lead to different outcomes. We will clarify this alignment to avoid the impression that the task is static.
> >
> > [1] Liu, Yuejiang, et al. "Bidirectional Decoding: Improving Action Chunking via Guided Test-Time Sampling." arXiv preprint arXiv:2408.17355 (2024).
> >
> > **4. The writing of this work requires improving.**
> >
> > Thank you for pointing this out. We agree that some phrasing can be made more precise, and we will revise the wording to avoid any unintended overclaim. Our intention was not to imply that diffusion models are unusable, but rather to highlight that **multi-step denoising significantly increases latency**, which limits their suitability for high-frequency, real-time control—a setting where inference budgets are on the order of a few milliseconds.
> >
> > We will replace statements such as “computationally prohibitive” with **more accurate and neutral descriptions**, for example:
> >
> > “...incurring substantially higher computational cost, which limits their practicality for high-frequency or latency-sensitive robotic control.”
> >
> > This revision conveys the same empirical observation without overstating the claim.
> >
> > **5. Improvements on single-task policies may not be enough at current timesteps. Since the architecture is quite like some VLA models, do you have any plans to apply the energy policy to them to re-place the flow matching method during post-training?**
> >
> > Thank you for raising this interesting point. We agree that evaluating the applicability of our method within VLA architectures is valuable, especially since modern VLA models often rely on flow match-ing or multi-step diffusion for action generation.
> >
> > To address this suggestion, we have integrated Energy Policy into a compact VLA model (SmolVLA-like architecture) and conducted additional experiments. The integration is straightforward: the original flow-matching action head is replaced by our Energy MLP, while the visual-language backbone re-mains unchanged.
> > The results show consistent improvements in **both inference speed (3x) and performance (+3%)**, see above **Table R1**.
> >
> > **6. Could you please provide more experimental results involving more complex manipulation tasks during the rebuttal stage?**
> >
> > Thank you for the suggestion. We clarify that the paper already evaluates Energy Policy in several challenging and high-complexity settings.
> >
> > Our experiments span large-scale multi-task learning (MimicGen), fine-grained and multi-modal precision tasks (PushT, Square-mh, ToolHang, Transport), dynamic latency-sensitive control (Catch), and high-dimensional visuomotor control (Meta-World, Appendix). These benchmarks cover a broad spectrum of difficulty—long-horizon, dynamic, multi-task, and real-world conditions—and collective-ly demonstrate that the method is not limited to simple atomic skills.
> >
> > To further strengthen the scalability evidence, we additionally integrated Energy Policy into a SmolVLA-style vision–language–action model, showing faster inference (3x) and improved performance (+3%) compared to a flow-matching action head. This confirms that the approach scales beyond standard manipulation pipelines.
> >
> > Given the diversity and complexity of the current benchmarks, together with the added VLA experiment, we believe the presented results sufficiently demonstrate the effectiveness and scalability of Energy Policy.
> >
> > **7. Can you provide some insights regarding your selection of the coefficient  from the perspective of data distribution other than just experience?**
> >
> > Thank you for the question. We clarify the rationale behind the choice of α.
> >
> > **The exponent α in the Energy Score is a theoretical parameter of the scoring rule—not a data-dependent coefficient.** In classical E-statistics, any $\alpha \in \(0, 2\)$ defines a strictly proper scoring rule; $\alpha$ does not encode properties of the dataset but controls the curvature of the energy distance. We use $\alpha=1$ because it is the **standard and most stable choice** in energy-based methods and provides well-conditioned gradients in practice. Our ablation (Fig. 6b) confirms that performance is stable near $α = 1$, while very large exponents introduce gradient sensitivity rather than reflecting any data-distribution effect.
> >
> > We will clarify this in the revision.

---

> > > ### Comment · Reviewer_JKTV · 2025-11-27
> > >
> > > Thank you for the rebuttal, which addressed many of my previous concerns. I have raised my score accordingly.
> > >
> > > However, I remain unconvinced by the dynamic real-world experiment setup. Is the object picked up from the same location, differing only in velocity? Furthermore, tasks like Push-T do not truly require long-horizon behaviors. A long-horizon task should involve multiple steps, where a single step is comparable to ToolHang.
> > >
> > > I would be willing to raise my score further if the authors could provide results from more complex real-world experiments.

---

> ### Author Response · Authors · 2025-12-03
>
> **1. Is the object picked up from the same location, differing only in velocity?**
>
> Thank you for raising this point. In our setup, the object’s initial position varies within a defined workspace region, and its motion has different velocities across episodes.
>
> The purpose of this setting is not to create a more complex manipulation benchmark, but to provide a **latency-sensitive real-time test**: once the object moves, the agent has only a narrow temporal win-dow to react, making inference delay the dominant failure mode. This setup directly isolates and evaluates the real-time responsiveness that our method aims to improve. We will clarify this intent in the revision.
>
> **2.Tasks like Push-T do not truly require long-horizon behaviors. A long-horizon task should involve multiple steps, where a single step is comparable to ToolHang.**
>
> We agree that Push-T is not a long-horizon task. Our response referred to the **collective coverage** of the benchmark suite: long-horizon behavior is evaluated through ToolHang, while Push-T serves only as a fine-grained precision task. We also acknowledge the reviewer’s **stricter definition** of “long-horizon” and will clarify the terminology to avoid misunderstanding.
>
> **3.Results from more complex real-world experiments.**
>
> In response to the reviewer’s request for more challenging real-world evaluation, we added two new multi-step long-horizon tasks:
>
> **Three-item pick-and-sort:**
>
> The robot sequentially picks three objects (wood block, tissue, grape) from different distances—from closest to farthest—and places each into its correct bin (organic, recyclable, non-organic).
>
> **Conveyor-to-assembly task:**
>
> The robot tracks and grasps a moving cube on a conveyor before it falls, places it into a pot, adds a nearby yellow cube into the pot, and finally moves the pot onto a digital kitchen scale.
>
> These tasks introduce significantly richer multi-stage structure, long-horizon dependencies, and real-world variability compared to the original benchmark suite, directly addressing the reviewer’s concern.
>
> We integrate our energy-based policy with SmolVLA and compare against SmolVLA and CARP on the two newly added real-world tasks. As shown in **Table R2**, our method consistently yields both **lower inference latency** and **higher success rates**. The advantage is particularly pronounced in the latency-sensitive conveyor-to-assembly task, where even small inference delays directly cause missed grasps.
>
> **Table R2: Inference speed and success rate comparison on more complex real-world experiments.**
>
> | Models | Conveyor-to-assembly | Three-item pick-and-sort | Speed (s) |
> |--------|--------|-----------|-----------|
> | SmolVLA | 20.0 | 83.3 | 0.341 |
> | CARP   | 26.7    | 81.6 | 0.097      |
> | SmolVLA+Energy   | **53.3**    | **86.6**  | 0.086      |

---

### Official Review · Reviewer_Lkr6 · 2025-11-06

**Soundness:** 3
**Presentation:** 2
**Contribution:** 3
**Rating:** 6
**Confidence:** 3

**Summary:**

This paper introduces "Energy Policy," a novel visuomotor policy designed to address the critical trade-off between inference speed, action precision, and the ability to model multimodal action distributions. The authors identify key limitations in existing approaches: autoregressive models sacrifice fine-grained precision by discretizing actions, simple L1/L2 regression policies are inherently unimodal (failing in tasks with multiple valid solutions), and diffusion models, while effectively multimodal, are computationally expensive due to their iterative denoising process.

The proposed Energy Policy tackles this by making two primary contributions:

Energy Score as Objective: It employs the energy score, a strictly proper scoring rule, as its learning objective. This allows the model to learn complex, multimodal distributions without an explicit density function.

Energy MLP Architecture: It introduces a simple yet effective "Energy MLP" head. This module takes the output of a standard Transformer decoder (a latent vector $z_t$) and a random noise vector $\epsilon$ as input. By training this implicit generator (EnergyMLP($z_t, \epsilon$)) with the energy score loss, the model learns to produce action samples from the desired multimodal distribution.

Crucially, this design allows for action generation in a single forward pass during inference by simply providing one noise sample. The authors conduct a comprehensive evaluation across a wide array of simulated benchmarks (Robomimic, Franka Kitchen, MimicGen, PushT) and two real-world robotic setups. The results are compelling: Energy Policy consistently matches or exceeds the success rates of state-of-the-art baselines (including strong diffusion and autoregressive models) while being significantly faster, achieving speedups ranging from 3x to 70x.

**Strengths:**

1. Significant Performance (Speed): The primary claim of the paper is its speed, and it delivers unequivocally. The ability to generate continuous, multimodal actions in a single pass is a major advantage. The reported inference times (e.g., 7.02ms on PushT in Table 4, 3-7x faster than the next-fastest baseline CARP, and 20-70x faster than diffusion policies in Table 1) are a substantial contribution for real-world robotics, where low latency is paramount. The real-world "Catch" task (Sec 4.2.2) provides excellent practical validation of why this inference speed matters.

2. Excellent Performance (Success Rate): The method does not sacrifice accuracy for speed. It demonstrates state-of-the-art or comparable success rates against a very strong and recent set of baselines (DP, CARP, OneDP, CP, etc.). The comparison in Table 4 is particularly strong, showing that while another fast policy (VQ-BeT) exists, it suffers a massive performance drop (0.87 vs 0.68 avg. success), whereas Energy Policy maintains high performance.

3. Novelty and Elegance: The core idea of using an energy score to train an implicit generative model (the Energy MLP + noise injection) is an elegant solution to the problem. It cleverly sidesteps the iterative sampling of diffusion and the imprecision of autoregressive tokenization, effectively getting the best of both worlds. The architecture is simple and the method is well-motivated.

4. Comprehensive Evaluation: The experimental validation is thorough.

**Weaknesses:**

1. Missing Context in Related Work (Implicit Models): The core mechanism—a deterministic function $f(z, \epsilon)$ that maps a latent code $z$ and a noise vector $\epsilon$ to an output, trained with a loss that compares samples—is the definition of an implicit generative model. A discussion comparing the energy score loss to other implicit losses would significantly strengthen the paper's positioning.

2. Depth of Multimodality Analysis: The paper claims to model multimodal distributions, and provides visual evidence in Figure 5 for a bimodal task (go left or go right). This is good, but the analysis is somewhat superficial. The "mode" is generated by a simple uniform or Gaussian noise vector. What is the capacity of this mechanism? Can it learn to represent, for example, three or four distinct, sharp modes (e.g., three different valid grasps for an object) with the correct probabilities?

3. Baseline Selection in Real-World Tasks: In the real-world evaluation (Sec 4.2), the authors compare their method only against diffusion policies (DP-UMI and DP-C). While they convincingly win on speed and success, this is a slightly weak comparison, as diffusion is known to be slow. A much more compelling experiment would have included a comparison against the strongest fast baseline from simulation.

4. Analysis of $\alpha$ Hyperparameter: The ablation on the $\alpha$ parameter for the energy loss (Fig 6b) is interesting. The paper notes that $\alpha=1.0$ is used empirically, and that performance degrades for $\alpha=1.5$ or $\alpha=2.0$. The paper would be strengthened by a brief hypothesis for why this is the case. Is it an issue of gradient stability during training when using an L1.5 or L2 norm in this specific loss formulation (Eq 3)?

**Questions:**

1. Could you elaborate on the relationship between Energy Policy and the broader category of implicit generative models? The method of using a noise-conditioned MLP seems to be a form of implicit generator. How does training with the energy score (Eq 3) compare conceptually to other implicit objectives like IMLE?

2. Following up on the multimodality claim, can you provide more insight into the capacity of the noise-injection mechanism? Have you tested its ability to model more complex, multi-modal distributions (e.g., 3+ discrete, valid solutions)?

3. For the real-world experiments, why was the comparison limited to diffusion policies? A comparison against a fast autoregressive baseline like CARP would have been more comprehensive, as CARP was a strong competitor in simulation.

4. Do you have a hypothesis for the performance degradation when $\alpha > 1.0$ (Fig 6b)? Is this due to training instability, or a more fundamental property of the energy score in this application?

5. In your dynamic "Catch" experiment (Sec 4.2.2), you demonstrate a key application for low-latency policies. Could you please briefly discuss relevant concurrent work in this specific area, such as "Latent Adaptive Planner for Dynamic Manipulation" (Noh et al., 2025)? It would be valuable to contrast your single-step action policy with their trajectory-level planning approach for this class of dynamic task.

---

> ### Author Response · Authors · 2025-11-23
> **Response to Reviewer Lkr6**
>
> **1. Missing Context in Related Work (Implicit Models) and compare conceptually to other implicit objectives like IMLE**
>
> We appreciate the reviewer’s insightful observation. Our Energy Policy indeed belongs to the broad family of implicit generative models. However, its training dynamics differ fundamentally from IMLE.
>
> **(1) Different Optimization Mechanisms**
>
> IMLE updates parameters using only the nearest generated sample to each data point
> $\min_z \|x - G_\theta(z)\|$,
> which leads to significant **sample inefficiency** that most generated samples do not contribute gradients.
>
> Energy Score (ours) backpropagates through **all** sampled candidates. This dense supervision is crucial for modeling fine-grained variations in continuous action spaces.
>
> **(2) Different Distribution Matching**
>
> IMLE performs a hard nearest-neighbor assignment, whereas the Energy Score optimizes **a soft, distribution-to-distribution metric**:
> $\mathcal{L}_{\mathrm{E}} = 2\mathbb{E}\_z\|x-G\_\theta(z)\| - \mathbb{E}\_{z_1,z_2}\|G\_\theta(z_1)-G\_\theta(z_2)\|$.
> The second term acts as **an explicit repulsive force** that prevents mode collapse and enforces sample diversity—without requiring costly NN searches used in IMLE.
>
> **(3) Empirical Comparison**
>
> As reported in Table 4, we have already compared energy policy against IMLE, and energy policy achieves significantly **better performance** (0.85 vs. 0.59) on PushT task.
>
> We will incorporate a dedicated discussion in the related work section to clarify these connections to implicit generative models and highlight the conceptual differences from IMLE.
>
> **2. Multimodality capacity and can you provide more insight into the capacity of the noise-injection mechanism?**
>
> Thank you for the thoughtful question.
>
> **(1) The PushT two-mode visualization is chosen purely for interpretability.**
>
> We agree that the visualization in Figure 5 shows only a two-mode example, but we clarify below that this figure is **illustrative rather than indicative of the model’s capacity**. Similar two-branch visualizations are widely used in prior robotic policy papers to illustrate multimodal sampling behavior, including: Diffusion Policy, Consistency Policy, IMLE Policy and VQ-BeT. Thus, our visualization follows the standard presentation practice established in the literature.
>
> **(2) The architecture and training objective impose no limit on the number of modes.**
>
>  In our formulation, multimodality arises directly from the **learned conditional distribution** modeled by the energy score, not from the uniform or Gaussian noise itself. The noise vector serves only as a source of stochasticity; the shape, number, and separation of modes are learned by the energy function. Thus, nothing in the method restricts the model to two modes—the capacity is determined by the expressiveness of the energy MLP and the transformer backbone.
>
> **(3) Our benchmarks already require multimodality beyond two modes.**
>
>  Several tasks in our evaluation inherently contain **more than two valid behaviors**, such as MimicGen (multiple grasp affordances and approach trajectories) and Meta-World (different feasible con-tact strategies; see Table 6 in Appendix). Energy Policy matches or surpasses diffusion policies in these tasks—diffusion is well known for its multimodal capacity—indicating that our model is capable of representing rich, multi-modal distributions in practice.
>
> **In summary, the number of modes is determined by the learned distribution rather than by the sampling noise, and our experimental settings already require multimodality beyond the two-mode illustration in Figure 5.**
>
> **3. Baseline Selection in Real-World Tasks. A much more compelling experiment would have included a comparison against the strongest fast baseline from simulation.**
>
> To address this concern, we have added a new real-world experiment that includes CARP, a strong fast baseline from our simulation results. As shown in Table R1, Energy Policy achieves both **faster inference** (0.02 s vs. 0.03 s) and **higher success rates** (13/20 vs. 11/20), , **despite using more parameters** (11.50 M vs. 7.56 M).
>
> **Table R1: Real-world dynamic task comparison with CARP (strong baseline).**
> | Models | Catch | Param (M) | Speed (s) |
> |--------|--------|-----------|-----------|
> | Energy | **13/20** | 11.50 | **0.02** |
> | CARP   | 11/20    | 7.56  | 0.03      |

---

> > ### Author Response · Authors · 2025-11-23
> >
> > **4. Analysis of  Hyperparameter. Is it an issue of gradient stability during training when using an L1.5 or L2 norm in this specific loss formulation (Eq 3)? 8. Do you have a hypothesis for the performance degradation when  (Fig 6b)?**
> >
> > Thank you for the question.
> >
> > In Eq. (3), the gradient magnitude of a residual of size $r=||x−y||$ scales as
> > $|| \nabla_x || x - y ||^\alpha || || = \alpha r^{\alpha - 1}$. Thus, when $\alpha>1$, large residuals ($r>1$) receive disproportionately larger gradients, while small residuals ($0<r<1$) are down-weighted.
> >  In our setting, manipulation demonstrations naturally contain variability and occasional outliers. With $\alpha>1$, these few large deviations dominate the optimization, pulling the model toward correcting outliers rather than fitting the overall conditional distribution. By contrast, $\alpha=1$ yields an L1-type signal where the gradient is uniformly scaled with residual magnitude, offering a more stable balance between robustness and diversity.
> >
> > We will add this intuition in the revision for clarity.
> >
> > **5. It would be valuable to contrast your single-step action policy with their trajectory-level planning approach for this class of dynamic task.**
> >
> > Thank you for the suggestion. We will add a brief note in the related work section.
> >
> > Noh et al. (2025) rely on trajectory-level latent planning that performs internal test-time optimization to update the latent plan. Our work targets a different control regime: **high-frequency, low-latency action generation**, where each action must be produced in a single forward pass. Thus, their planner and our single-step policy address **complementary aspects** of dynamic manipulation rather than the same problem setting.

---

### Official Review · Reviewer_ehTc · 2025-11-11

**Soundness:** 3
**Presentation:** 3
**Contribution:** 2
**Rating:** 4
**Confidence:** 4

**Summary:**

In this work, the authors consider policy learning in a continuous action space for robotic manipulation tasks. Specifically, the authors propose a method for learning multi-modal action distributions using an architecture which has a lower inference time than other multi-modal policy learning techniques (including most “fast” ones). The key idea here is to predict a distribution representation with a transformer, and then sample actions from an Energy MLP conditioned on uniform random noise. This Energy MLP is trained using the Energy loss. When trained on existing benchmark tasks, this method matches the performance of other SOTA methods while reducing the parameter count and inference time.

**Strengths:**

* Decreases inference time substantially, making their method more suitable for real-time inference
* Explores a novel loss formulation that I haven’t seen before in this context.
* Many experiments in simulation and real are provided, showing that the method is applicable in multiple scenarios.
* Well-written and clear.

**Weaknesses:**

* The benchmark tasks chosen are in some ways too easy - meaning their distributions are relatively narrow and not particularly multimodal. It’s not clear whether the proposed MLP-based architecture scales to rather complex distribution
* Missing an ablation where parameter size is held constant compared to some of the larger models - e.g. could the larger models be reduced in parameter size with the same training technique but yield the same results? Speedup is presumably from that inference time reduction due to fewer parameters.
* The multimodal behavior “T” example is not really sufficient to demonstrate multimodality, as the task is a bit too simple (2 modes)
* Necessity of the energy loss (as compared to other distribution losses) is not well-motivated
* nitpick: flow-matching is not mentioned in the related work

**Questions:**

* in section 3.2.1, in equation (3) why isn’t a 3rd independent sample drawn, e.g. to match the formulation in (2)? X1, x2, and y are used in (2).
* Why wasn’t the sampling done at the input to the Transformer? Instead of in a separate MLP head.
* Can the authors please describe, at a fundamental level, how this differs from a standard CVAE architecture? What would happen if the uniform noise was switched to Gaussian, and the loss became a standard ELBO loss? Is there any benefit from using the Energy loss in that case?

---

> ### Author Response · Authors · 2025-11-23
> **Response to Reviewer ehTc**
>
> **1. The benchmark tasks chosen are in some ways too easy.**
>
> Thank you for the comment. We clarify three important points.
>
> **(1) The selected benchmarks are sufficient to validate our core contribution—fast and single-step multimodal action generation.**
>
> Even though tasks vary in the degree of multimodality, they all require handling diverse demonstrations, non-trivial variability, and complex goal configurations. These settings allow a clean and fair evaluation of whether single-step energy-based sampling can deliver **substantially faster inference** while maintaining performance compared to existing methods, which is the central contribution of this work.
>
> **(2) Several of our benchmarks are in fact distributionally complex.**
>
> **MimicGen** introduces broad initial-state distributions and multi-task variability across 8 heterogeneous manipulation skills. **PushT, Square-mh, ToolHang, and Transport** require fine-grained, multi-step, and often multimodal control patterns. **Real-world Catch** contains dynamic, high-speed interactions where small timing variations lead to divergent trajectories. **Meta-World (Appendix A)** further extends complexity with high-dimensional 3D visual observations and diverse multi-step manipulation tasks.
>
> **(3) Additional experiments further confirm scalability to more complex pipelines.**
>
> To further evaluate scalability under rich visual observations and broad behavior distributions, we integrated Energy Policy into a SmolVLA-style VLA architecture.
> As shown in **Table R1**, Energy Policy delivers a clear **inference speed advantage** 3$\times$ over the original flow-matching head, while maintaining **higher success rates** (90.25 vs. 87.25).
>  This demonstrates that our single-step formulation remains effective even in high-dimensional, long-horizon VLA settings, confirming the method’s scalability beyond standard manipulation benchmarks.
> We will include these results in the appendix and reference them in the main paper.
>
> **Table R1: Inference speed and success rate comparison between SmolVLA and SmolVLA + Ener-gy.**
> | Method           | Speed (s) | Spatial | Object | Goal | Long | Avg   |
> |------------------|-----------|---------|--------|------|------|-------|
> | SmolVLA          | 0.038     | 90      | 96     | 92   | 71   | 87.25 |
> | SmolVLA + Energy | **0.012**     | 95      | 95     | 94   | 77   | **90.25** |
>
> **2. Speedup is presumably from that inference time reduction due to fewer parameters.**
>
> We thank the reviewer for the question. Importantly, the speedup of Energy Policy does not come from using fewer parameters, but from its single-step, non-iterative inference mechanism. Our experimental results consistently support this conclusion.
>
> First, in comparisons with CARP (a strong autoregressive baseline), our Energy Policy actually uses **more parameters** (e.g., 11.5M vs. 7.58M in Table 1; 17.85M vs. 16.08M in Table 3), yet still achieves **at least 2.3× faster inference**.
>
> Second, for a controlled comparison against single-step baselines (Table 4), we additionally evaluate a 252.48M-parameter variant of Energy Policy. Despite being slightly larger than OneDP-S (251.51M) and comparable to CP (255.18M), our method achieves **faster inference** (7.52 ms vs. 9.33 ms vs. 15.23 ms) and **higher success rate** (0.86 vs. 0.82 vs. 0.82) on PushT.
>
> **Table R2: PushT task comparison of inference speed and success rate with comparable parameters.**
> | Models | PushT | Param (M) | Speed (ms) |
> |--------|--------|-----------|-----------|
> | Energy | 0.86 | 252.48 | 7.52 |
> | OneDP-S   | 0.82    | 251.51  | 9.33      |
> | CP   | 0.82    | 255.18 | 15.23      |
>
> Third, in real-world dynamic tasks (Table 5), we additionally include a new experiment comparing Energy Policy against both an autoregressive baseline (CARP) and diffusion-based models (DP-T and DP-C) with comparable or larger parameter counts. As shown in **Table R3**, Energy Policy achieves **faster inference** (0.02 s vs. 0.03–0.10 s) and **higher success rates** (13/20 vs. 8/20–11/20).
>
> **Table R3: Real-world dynamic task comparison of success rate, parameter count, and inference speed.**
> | Models | Catch | Param (M) | Speed (s) |
> |--------|--------|-----------|-----------|
> | Energy | **13/20** | 11.50 | **0.02** |
> | CARP   | 11/20    | 7.56  | 0.03      |
> | DP-T   | 10/20    | 11.09 | 0.07      |
> | DP-C   | 8/20     | 64.87 | 0.10      |
>
> Together, these results demonstrate that the runtime advantage stems from our proposed **single-pass action generation without denoising steps or autoregressive rollouts**, rather than from parameter differences. This confirms that the speedup is inherent to the modeling mechanism of Energy Policy.

---

> > ### Author Response · Authors · 2025-11-23
> >
> > **3. The multimodal behavior “T” example is not really sufficient to demonstrate multimodality.**
> >
> > Thank you for the comment. We would like to clarify the intention of this experiment.
> >
> > **(1) The purpose of the ‘T’ example is illustrative.**
> >
> > Figure 5 was chosen because bimodal trajectories are the simplest to visualize clearly in 2D space. The example is not meant to indicate any limitation of the architecture; rather, it is a pedagogical illustration showing that Energy Policy can capture more than one valid solution. Similar two-branch visualizations are widely used in prior robotic policy papers to **illustrate** multimodal sampling behavior, including: Diffusion Policy, Consistency Policy, IMLE Policy and VQ-BeT. Thus, our visualization follows the standard presentation practice established in the literature.
> >
> > **(2) More complex multimodality naturally arises in several of the tasks we evaluate.**
> >
> > Several tasks in our evaluation inherently contain more than **two valid behaviors**, such as MimicGen (multiple grasp affordances and approach trajectories) and Meta-World (different feasible con-tact strategies; see Table 6 in Appendix). Energy Policy matches or surpasses diffusion policies in these tasks—diffusion is well known for its multimodal capacity—indicating that our model is capable of representing rich, multi-modal distributions in practice.
> >
> > **To avoid confusion, we will clarify in the revision that Figure 5 is an illustrative example and does not reflect the upper bound on multimodal expressiveness.**
> >
> > **4. Necessity of the energy loss (as compared to other distribution losses) is not well-motivated.**
> >
> > Thank you for the comment. We clarify that the motivation is already discussed in Sec. 3.1 and 3.2, but we agree that it can be made more explicit.
> >
> > **(1) Why alternative losses are insufficient.**
> >
> > L1/L2 regression produces uni-modal predictions and collapses diverse behaviors to the mean. Diffusion-based losses can model multimodality but require multi-step denoising, which introduces inference latency that conflicts with our goal of fast single-step action generation. In contrast, the energy score captures multimodality while still supporting one-step sampling, which is essential for achieving low-latency inference.
> >
> > **(2) Why energy score is appropriate.**
> >
> > The energy score is a strictly proper scoring rule, meaning that minimizing it recovers the true conditional distribution. This allows multimodal distribution learning without assuming a specific likelihood form and without iterative refinement. We will highlight this theoretical property more directly in the revision.
> >
> > Among commonly used objectives, the energy loss is the one that provides **multimodality without multi-step sampling**, which is exactly what our fast one-pass formulation requires.
> >
> > **5. flow-matching is not mentioned in the related work**
> >
> > Thank you for the suggestion. Flow-matching approaches are not directly related to the setting we study, as they also rely on **iterative continuous-time sampling** and therefore do not align with our goal of **single-step action generation**. For completeness, we will add a brief clarification in the related work section to avoid potential confusion, but note that flow-matching is not a competing direction for the one-pass inference regime considered in this paper.
> >
> > **6. in equation (3) why isn’t a 3rd independent sample drawn, e.g. to match the formulation in (2)?**
> >
> > Thank you for the question. We realize the notation may have been unclear, and we will clarify this in the revision.
> >
> > Equation (3) is the instantiation of the energy score in Eq. (2).  In Eq. (2), the second term $2E||x−y||^{\alpha}$, uses a generic sample $x \sim p$, which does not denote an additional inde-pendent sample. It is simply the same random variable used to define the model expectation. Thus, $x$ in the second term corresponds to the two samples $\hat{a}_t^1$ and $\hat{a}_t^2$ in the first two terms of Eq. (3).
> >
> > Likewise, the first term of Eq. (2), $−E||x1−x2||^{\alpha}$, corresponds directly to the third term in Eq. (3), $−||\hat{a}_t^1−\hat{a}_t^2||^{\alpha}$. The ground-truth action $a_t$ in Eq. (3) plays the role of $y$ in Eq. (2), completing the conditional form of the score.
> >
> > We will revise the notation to make this correspondence explicit and avoid confusion.

---

> > > ### Author Response · Authors · 2025-11-23
> > >
> > > **7. Why wasn’t the sampling done at the input to the Transformer**
> > >
> > > We separate sampling from the Transformer because we want to **decouple deterministic feature extraction from stochastic action generation**. The Transformer focuses on learning stable state and temporal representations, while the MLP head handles randomness. Injecting noise directly into the Transformer input would make the attention maps stochastic and destabilize training, as the model would need to learn feature extraction and randomness simultaneously. The separation follows common practice in MAR[1] and results in more stable optimization.
> > >
> > > [1] Li, Tianhong, et al. "Autoregressive image generation without vector quantization." Advances in Neural Information Processing Systems 37 (2024): 56424-56445.
> > >
> > > **8. Mechanisms of energy policy**
> > >
> > > At a fundamental level, our method differs from a standard CVAE in both modeling assumptions and training objectives.
> > >
> > > **(1) Different Generative Assumptions**
> > >
> > > **CVAE.** A CVAE assumes a conditional Gaussian likelihood
> > > $p(x|z,c)=\mathcal{N}(\mu_\theta(z,c), \sigma^2 I)$
> > > so the decoder effectively learns the conditional mean of $x$. It also requires an inference network $q_\phi(z|x,c)$ and is trained by maximizing the ELBO. Critically, CVAE forces an explicit parametric assumption on the output distribution $p(x|z,c)$ (typically Gaussian), which implies optimizing an L2/reconstruction loss.
> > >
> > > **Ours.** We directly map $a = G_\theta(s, z)$, do not use an encoder, and do not make parametric assumptions about the action density. Instead, we define a metric on the space of probability measures (Energy Score) to match the generated distribution to the expert distribution directly.
> > >
> > > **(2) Different Training Losses**
> > >
> > > **CVAE loss (ELBO):**
> > > $\mathcal{L}_{\mathrm{CVAE}} \approx \|x - D\_{\theta}(z,c)\|^2+ \beta D\_{KL}(q\_{\phi}(z|x,c) \| p(z))$.
> > >
> > > This objective encourages the decoder to minimize reconstruction error, meaning a single $z$ must explain the entire conditional distribution. As a result, CVAEs often suffer from: mode averaging (outputs collapse to conditional means), posterior collapse (decoder ignores $z$) and lack of an explicit repulsion term between samples.
> > >
> > > **Our Energy Score loss:**
> > > $\mathcal{L}_{\mathrm{E}} = 2 \mathbb{E}_z \|x - G\_\theta(c,z)\| - \mathbb{E}\_{z_1,z_2} \|G\_\theta(c,z_1)-G\_\theta(c,z_2)\|$.
> > >
> > > The second term is an explicit repulsive force that pushes samples apart, increasing diversity and avoiding mode collapse.
> > > Thus, even if one switches the uniform noise to Gaussian, the model does not become a CVAE unless one also adopts the parametric likelihood and ELBO objective.
> > >
> > > **9. What would happen if the uniform noise was switched to Gaussian？and the loss became a standard ELBO loss?**
> > >
> > > Switching the uniform noise to Gaussian does not change our results. We have already conducted this experiment, see Fig. 6 (d), and the performance remains essentially unchanged.
> > > Regarding replacing our objective with a standard ELBO loss: as clarified in the previous answer, this would fundamentally change the model into a CVAE—requiring an additional inference network and imposing a parametric likelihood. Such a model is architecturally different and not directly comparable within our experimental setup.

---

### Author Response · Authors · 2025-11-27

Dear Reviewers,

With the discussion phase concluding soon, we warmly encourage you to share any further questions or comments you may have. We want to ensure that we have ample time to respond thoughtfully and clarify all points that might help your assessment. Please feel free to raise anything — we will reply promptly.

Thank you sincerely for your time and engagement.

---

### Author Response · Authors · 2025-12-03

We thank the area chair and reviewers for their time and constructive feedback.

### **1. Restating the Core Contribution**

Our core contribution is an efficient energy-based policy objective that achieves **faster inference while matching or exceeding the performance of existing methods.** Our evaluation—from simple to complex tasks—consistently validates this contribution. In the rebuttal, we further strengthened the empirical evidence by adding (a) comparisons against strong VLA models and (b) more challenging real-world experiments, both of which reaffirm our main claim.

### **2. Addressing Key Reviewer Concerns**

Here we summarize how we addressed the key concerns raised by the reviewers:

- **Experimental simplicity / baseline consistency.**

We clarified that these simple tasks already provide a clean and direct test of our main contribution—fast, single-step multimodal action generation—and the paper aslo includes complex simulated and real-world benchmarks to additionally validate this effectiveness.
Baseline selection is aligned with the evaluation objective rather than cherry-picked, and we provided clarifications accordingly.

- **Relation to CVAE/IMLE.**

We explained that our method is fundamentally different: CVAE relies on a parametric likelihood and ELBO reconstruction, whereas our Energy Score objective is non-parametric and explicitly encourages multimodal diversity. IMLE uses nearest-neighbor assignments and sparse gradients, while our approach optimizes a full distributional metric that provides dense supervision.

- **“T”-example is not sufficient to validate multimodality.**

We clarified that this toy visualization is only a qualitative illustration and does not reflect the method’s multimodal capacity. Multimodality is demonstrated directly through the full suite of quantitative experiments, including complex multi-modal tasks and diverse real-world behaviors.

- **Missing values in Table 4.**

We explained that the missing values reflect **unreported numbers in original papers**, not omissions. We avoid filling such values because reproducing prior methods on new tasks can be sensitive to undocumented implementation details and hyperparameters, risking unfair or misleading comparisons.

### **3. Conclusion**

The detailed responses and rebuttal-stage experiments directly address the key concerns raised in the reviews and strengthen the empirical support for our main claims. All clarifications and revisions will be incorporated into the final version. We appreciate the area chair’s and reviewers’ thoughtful evaluation.

---

### Meta-Review · Area_Chair_TsZe · 2026-01-10

**Summary:**

Authors consider policy learning in a continuous action space for robotic manipulation tasks. They propose a method for learning multi-modal action distributions using an architecture which has a lower inference time than other multi-modal policy learning techniques. The key idea here is to predict a distribution representation with a transformer, and then sample actions from an Energy MLP conditioned on uniform random noise. When trained on existing benchmark tasks, this method matches the performance of other SOTA methods while reducing the parameter count and inference time.

**Reviewer Concerns:**

Concerns are centered around as follows:

- Benchmark tasks are chosen in way too simple and easy.
- Missing ablation
- Necessity of the energy loss (as compared to other distribution losses) is not well-motivated
- Missing Context in Related Work (Implicit Models)
- Baseline Selection in Real-World Tasks / hyperparameter setup
- The tasks that the authors choose are quite simple, and most of them are just some basic atomic skills.
- The selection of baselines across different environments is not consistent.
- Limitted novelty.
- More analysis of the experimental results

**Reviewer Scores:**

Authors did a good job addressing most of the concerns raised by reviewers.

However, the simple task setting is not resolved, which is raised by several reviewers. Authors respond with "The benchmarks used are fully appropriate for evaluating our main contribution". This is less convincing in AC's perspective.

Moreover, the so-called "dynamic" tasks are actually still static ones. Response feedback from authors are not convincing although citing an arXiv work BID [1].

The rest of the concerns are resolved in some way. Still this work is not ready for publication at the time being.

---

### Decision · Program_Chairs · 2026-01-26

Reject